# The Ins and Outs of RAS Effector Complexes

**DOI:** 10.3390/biom11020236

**Published:** 2021-02-07

**Authors:** Christina Kiel, David Matallanas, Walter Kolch

**Affiliations:** 1Systems Biology Ireland, School of Medicine, University College Dublin, Dublin 4, Ireland; christina.kiel@ucd.ie (C.K.); david.gomez@ucd.ie (D.M.); 2UCD Charles Institute of Dermatology, School of Medicine, University College Dublin, Dublin 4, Ireland; 3Conway Institute of Biomolecular & Biomedical Research, University College Dublin, Dublin 4, Ireland

**Keywords:** RAS oncogene, RAS signaling networks, RAS in human cancer, targeting RAS, computational modeling, personalized therapies

## Abstract

RAS oncogenes are among the most commonly mutated proteins in human cancers. They regulate a wide range of effector pathways that control cell proliferation, survival, differentiation, migration and metabolic status. Including aberrations in these pathways, RAS-dependent signaling is altered in more than half of human cancers. Targeting mutant RAS proteins and their downstream oncogenic signaling pathways has been elusive. However, recent results comprising detailed molecular studies, large scale omics studies and computational modeling have painted a new and more comprehensive portrait of RAS signaling that helps us to understand the intricacies of RAS, how its physiological and pathophysiological functions are regulated, and how we can target them. Here, we review these efforts particularly trying to relate the detailed mechanistic studies with global functional studies. We highlight the importance of computational modeling and data integration to derive an actionable understanding of RAS signaling that will allow us to design new mechanism-based therapies for RAS mutated cancers.

## 1. Introduction

Since the discovery of retroviruses transducing RAS oncogenes in 1964 [1], huge research efforts have been spent on understanding what RAS does [2]. There is a good reason for that. RAS proteins are mutated in 19% of all human cancers, and several prevalent and deadly cancers, such as colorectal, lung, and pancreatic cancers and metastatic melanoma are driven by RAS mutations [3]. Of the three RAS family members, KRAS is the most frequently mutated form in pancreatic, colorectal, and lung cancer. NRAS is preferentially mutated in melanoma and acute myeloid leukaemias, while HRAS mutations are rare [3]. In addition, germline mutations that activate RAS are responsible for RASopathies, a group of rare developmental disorders [4,5]. The oncogenic RAS mutations compromise the ability of RAS proteins to hydrolyze guanosine triphosphate (GTP) to guanosine diphosphate (GDP) or enhance the replacement of GDP by GTP. Physiologically, GTP binding is stimulated by Guanine nucleotide Exchange Factors (GEFs), while GTP hydrolysis is accelerated by GTPase Activating Proteins (GAPs). In the GTP bound form, RAS switches into a conformation where it can bind and activate effector proteins which transduce a wide range of signals that regulate almost any aspect of cellular physiology [6]. The known and well-characterized RAS signaling network is shown in Figure 1. Upon GTP hydrolysis, RAS proteins release their effectors, thereby terminating the signal. Most often RAS proteins are intermediates between surface receptors and intracellular signaling pathways that regulate proliferation, differentiation, survival, cell-cell interactions and migration [6]. The oncogenic mutations are thought to lock RAS in the active conformation, causing constitutive signaling through these downstream effector pathways, most prominently the RAF and phosphoinositide-3 kinase (PI3K) pathways [6]. In fact, when one includes mutations in these two RAS effector pathways, oncogenic activation of the RAS pathway may be involved in more than half of all human cancers [7]. Efforts to target oncogenic RAS signaling have been thwarted by the intricacies of RAS regulation and an ever expanding network of effectors that participate in mediating its oncogenic effects [8]. The take-home lesson was that we need to understand RAS signaling in much greater detail, if we want to successfully target it. In this review, we summarize the current knowledge of how RAS interacts with its effectors focusing on modalities that affect direct interactions as well as RAS isoform and tissue-specific interactions.

## 2. RAS Interactions with Effector Proteins

### 2.1. RAS Effector Proteins Bind to the ‘Switch Regions’ of RAS∙GTP

RAS effectors are specific for the GTP-bound conformation of RAS, where two structural regions—the so called ‘switch regions’ (switch I and switch II)—are in a ‘closed’ conformation, which exposes interface residues involved in binding to the RBD [9,10,11,12]. Once the γ phosphate is hydrolyzed, RAS∙GDP acts like a ‘loaded spring’ and releases switch I and II in ‘open’ conformations. While the molecular switch function of RAS builds on the efficient interaction of effectors in the GTP-bound state, an interaction with the GDP-bound form is not strictly impossible. Indeed, through structure-based mutagenesis of the CRAF-RAS binding Domain (RBD), it was possible to engineer mutants with high affinity for RAS∙GDP that induced CRAF-mediated pathway activity [13]. Importantly, the 3D structure of such a high-affinity RAS∙GDP-effector complex showed a similar closed conformation of the switch I/II regions, suggesting that the conformation of RAS in that region rather than the nucleotide state itself defines the ability to interact with effector [14].

### 2.2. Competition for Binding to RAS∙GTP Generates Different RAS-Effector Complexes

RAS effectors are defined as proteins containing at least one domain that can directly bind to RAS∙GTP—a domain with a ubiquitin-like topology [15,16]. While, on the basis of different consensus amino acid sequences, three different domain types exist that can bind to RAS oncoproteins, the RBD (SMART accession number: SM00455), the RA (SM00314), and the PI3K_rbd (SM00144), as their overall 3D structure is similar, we collectively refer to these domains as RBDs here. Based on sequence similarities and domain prediction tools, the human genome is predicted to contain 56 RBD-containing effector proteins, which fall into 12 effector pathways [17]. Noteworthy, the binding affinities of individual effector RBDs in complex with RAS∙GTP measured either in vitro using biophysical techniques [18,19,20] or calculated in silico using 3D structural modeling and energy calculations [21,22] span a wide range from low nanomolar to high micromolar K_d_ values [17] (Figure 2A). As only a handful of interface residues (‘hot spots’) define the strength of binding, it is not surprising that domains with a similar fold can indeed exhibit different binding affinities in complex with RAS∙GTP. Noteworthy, there is no relation between binding affinity and type of the domain binding to Ras (RBD, RA, or PI3K_rbd). Of the 56 effectors, eight are high-affinity binders (K_d_ values ≤ 1 μM), 35 are low-affinity binders (K_d_ values > 1 μM), four are likely not to bind to RAS∙GTP via their RBD, and for nine effectors no in vitro or in silico binding information is available.

Because all effector RBDs recognize the same switch regions of RAS∙GTP, binding of effectors is mutually exclusive. Hence, if the concentration of RAS∙GTP is limiting, effectors compete for binding to RAS [24,25,26]. Indeed, we have shown previously that increasing the concentration of the effector RIN1 in cultured cells decreased the phosphorylation of CRAF and its downstream targets, which was enhanced even further when RIN1 was artificially localized to the plasma membrane (PM) using a CAAX box [24]. Another consequence of the competition of effectors for binding to RAS is that increasing the levels of active RAS∙GTP, such as in the case of cancer mutants that prevent efficient hydrolysis of RAS∙GTP, is predicted to not only increase the overall amount of effectors in complex with RAS (quantitative change), but also qualitatively change the binding profile, where low-affinity effectors proportionally engage more with the additional amount of RAS∙GTP available [17,25]. Thus, understanding RAS-effector signaling requires studying all effectors present in a biological system, as the cellular output depends on the concentrations (and affinities) of all players present.

Our recent investigation using quantitative computational modeling of RAS signaling complexes in colon tissue [17] suggested that at physiologically low RAS GTP levels high-affinity RAS-RAF complexes dominate. Raising RAS GTP concentrations disproportionally increased low-affinity RAS effector complexes that regulate cell adhesion and epithelial barrier functions. These results indicate that RAS signaling changes not only quantitatively but also qualitatively depending on the level of RAS activation and the concentrations of different RAS effector proteins. Extending this work to include protein abundances measured in 29 human tissues [27], we predicted that only nine of the 56 effectors form significant complexes with RAS∙GTP in at least one tissue (with a threshold >5% of RAS∙GTP in complex with a particular effector) [23] (Figure 2B). Of those, five effectors (ARAF, BRAF, CRAF, RGL2, and RASSF5) are predicted to form complexes in all/most of the 29 tissues, and four (RALGDS, AFDN, SNX27, and RASSF7) only in some tissues. However, there is some change in the amount of individual RAS-effector complexes and in the ranking among these nine effectors, confirming that tissue-specific RAS and effector abundances (together with affinity) can alter effector complexes. Hence, tissue-specific RAS signaling functions can at least in part be explained by the combination of abundance and affinity of the effectors expressed in different tissues.

### 2.3. Modulation of RAS Effector Recruitment

#### 2.3.1. Protein Domains that Facilitate Effector Recruitment to RAS

Approximately half of the RAS effectors reported in the literature have low-affinity RBDs (Figure 2A). However, some of these low-affinity effectors are well established and important RAS effectors, e.g., phosphoinositide-3 kinase (PI3K) [28]. This poses the question whether there are additional ways to enhance the interactions between RAS and its effectors. A simple, yet effective mechanism is to recruit effectors into the vicinity of RAS. Indeed, our analysis of tissue-specific RAS effector complexes also revealed that 32 out of 56 effectors could only bind to RAS∙GTP in significant amounts if they were recruited to the membrane via additional domains [23]. As RAS proteins are anchored to membranes, a domain that directs a potential RAS effector protein to a membrane increases the affinity between RAS and this effector by >100 fold, simply by reducing a three-dimensional reaction space to two dimensions [29] (Figure 3A).

RAS effectors are (generally) larger and multi-domain proteins, and one possibility for prolonging the residence time of effectors once recruited to RAS∙GTP is their trapping in the actin meshwork (Figure 3B). The cell cortex, a mesh-like structure composed of actin microfilament and over a hundred actin-binding proteins, is connected to the cytosolic side of the PM and controls cell morphogenesis [30]. In addition, the actin meshwork impacts signaling events at the PM, in particular by limiting the diffusion of proteins in vicinity of the PM, such as RAS and effector proteins [31,32,33]. Indeed, disruption of the action meshwork prevents the formation of KRAS nanoclusters in the PM [33] and enhances the mobility of full length CRAF at the PM [34]. While the actin meshwork is likely to cause prolonged residence times for all effectors once they are recruited to RAS∙GTP via their RBD, effectors may also directly bind to proteins localized in the actin meshwork. Information obtained from the HuRI database that stores direct binary protein-protein interactions obtained from genome-wide yeast two hybrid experiments [35], reports direct physical interactions between four effectors and interactors localized to the actin cytoskeleton (based on SysGO [36]) (Figure 3B). These binary interactions are between (i) the effector RIN1 and CIP4/TRIP10, a CDC42 interacting protein [37]; (ii) the effector RASSF10 and TNNI3, a cardiac muscle protein that is an inhibitory subunit of troponin [38]; (iii) the effector RAPGEF4 and DEF6, a guanine nucleotide exchange factor for Rho GTPases [39]; and (iv) the effector GRB10 and KANK2, involved in actin stress fibers formation through the regulation of Rho signaling [40].

Many RAS effectors, in addition to their RBD, contain domains with affinity to membranes including the PM or to proteins localized at the PM [17]. Indeed, 28 RAS effectors contain membrane-binding domains (Figure 3C). These comprise:C1 domains (protein kinase C conserved region 1 domains; SMART accession number: SM00109); usually bind to phorbol esters causing hydrophobic residues being exposed, which enables binding to the PM [41]. These domains are present in the effectors ARAF, BRAF, CRAF, RASSF1, RASSF5, MYO9A, MYO9B, and DGKQ.PX domains (PhoX homologous domains; SM000312) bind phosphoinositides with varying lipid-binding specificities [42,43]. The effectors PIK3C2B, PIK3C2G, PIK3C2A, and SNX27 contain PX domains.C2 domains (protein kinase C conserved region 2 domains; SM00239) belong to a large family of domains with a shared overall fold, yet broad functional properties—ranging from Ca^2+^ binding to multiple lipid-to-protein interactions [44]. The effectors PIK3C2B, PIK3C2G, PIK3C2A, and PLCE1 contain C2 domains.PH domains (pleckstrin homology domains; SM00233) belong to one of the most abundant domain families in the human genome; they bind to phosphoinositides within the PM [45,46]. One or more PH domains are present in the effectors TIAM1 (two domains), TIAM2 (two domains), ARHGAP20, ARAP1 (five domains), ARAP2 (five domains), ARAP3 (five domains), APBB1IP, RAPH1, MYO10 (two domains), GRB7, GRB10, and GRB14.B41 domains (Band 4.1 homologues domains; SM00295) are involved in both localizing proteins to the PM and in providing structural and regulatory functions [47]. B41-domain containing effectors are KRIT1 and MYO10.DEP domains (domain found in disheveled, egl-10, and pleckstrin domains; SM00049) function in the recruitment of proteins to PMs [48], and effectors containing DEP domains are RAPGEF3 and RAPGEF4.

In addition to domains directly binding to the PM, effectors often contain typical domains involved in intermolecular protein-protein interactions, such as SH2 (SMART accession number: SM00252), SH3 (SM00326), PDZ (SM00228), PTB (SM00462), ANK (SM00248), and FHA (SM00240) domains [17]. Information obtained from the HuRI database [35] reports direct binary interactions between nine effectors and interactors localized to the PM (based on SysGO [36]) (Figure 3D). Those effectors are RIN1 (binding to PICK1, ANKS1B), RASSF10 (binding to GNG13 and FAM171A1), RASSF3 (binding to TFRC and ITPRIP), RASSF8 (binding to FRMD6), RGS12 (binding to GNAI1 and GNAI3), GRB7 (binding to PICK1, TGM5, LAX1, ERBB2, ERBB3, KIT, RET, and INSR), GRB10 (binding to PICK1, BLK, IGF1R, RET, INSR, EGFR, and KIT), and GRB14 (binding to FGFR1, and INSR). In addition, the class I PI3K effectors PIK3CA, PIK3CB, PIK3CD, and PIK3CG can be recruited to the PM via their regulatory subunits, which use their SH2 domains to bind to receptors [49], such as EGFR, ERBB2, ERBB3, KIT, PDGFRA, and PDGFRB [35].

Information for each of the 56 effectors in the 12 RAS effector groups is summarized in Table 1 and Figure 4.

#### 2.3.2. Posttranslational Modifications That Regulate RAS Interactions with Its Effector Proteins

Posttranslational modifications (PTMs) are well known to regulate protein–protein interactions either by serving as docking sites, by physically blocking interactions, or by inducing conformational changes that modify affinities. All three RAS proteins can be ubiquitinated at different sites regulating RAS stability, subcellular localization and effector interactions [66] (Figure 5A–D). Mono-ubiquitination of lysine 117 in HRAS [67] and lysine 147 in KRAS [68,69] increases RAS GTP loading and interaction with RAF, PI3K and RALGDS effectors (Figure 5A). Interestingly, HRAS mono-ubiquitination increases GTP loading independent of GEFs [67], while KRAS mono-ubiquitination reduces the interaction with GAPs [68], both resulting in the activation of RAS and downstream pathways. Interestingly, mono-ubiquitination also can impair the constitutive lysosomal degradation of KRAS entertained by the β-TrCP1 ubiquitin ligase complex (Figure 5B). In this pathway the mono-ubiquitination targets a regulator of this KRAS degradation complex [70]. The ubiquitin protein ligase Smurf2 mono-ubiquitinates the ubiquitin-conjugating enzyme UBCH5 to form a complex that poly ubiquitinates β-TrCP marking it for degradation. Thus, KRAS is stabilized because its degradation complex is destroyed. Indeed, knocking down Smurf2 reduces KRAS abundance, impairs clonogenic survival and prolongs tumor latency in mutant KRAS-driven lung and colorectal tumor models [70].

Di-ubiquitination of HRAS and NRAS on unknown sites promotes RAS endocytosis (Figure 5C). Although these modifications do not affect the binding of RAF proteins, they diminish RAF activation by shortening the dwell time of RAF at the PM and its exposure to activating events [71]. Interestingly, this modification is mediated by the E3 ubiquitin ligase Rabex-5 in a RIN1 dependent manner [72]. As RIN1 is a RAS effector, this circuitry may constitute a negative feedback loop that limits RAS activation. Similarly, Rabex-5 mediated RAS ubiquitination requires tyrosine phosphorylation at the very N-terminus [73], suggesting that the activation of tyrosine kinases is part of this negative feedback loop.

Poly ubiquitination of HRAS, NRAS and KRAS on lysine 170 induce their degradation by the proteasome and reduction of ERK signaling (Figure 5D). This route is mediated by the β-TrCP1 ubiquitin ligase that is recruited to RAS by the adaptor protein LZTR1 [74,75,76]. It also may constitute an interface for crosstalk with the WNT pathway, which promotes oncogenic transformation by stabilizing β-catenin and allowing the formation of β-catenin/TCF transcription factor complexes [77]. The WNT pathway kinase responsible for marking β-catenin for degradation is GSK3β. WNT signaling dismantles the GSK3β kinase complex preventing β-catenin phosphorylation and degradation [77]. GSK3β can phosphorylate threonines 144 and 148 in HRAS triggering its poly ubiquitination and subsequent degradation [78]. This phosphorylation requires prior β-catenin degradation, as stabilized β-catenin can bind to HRAS and KRAS physically obstructing the GSK3β phosphorylation sites and preventing RAS degradation [79]. Thus, WNT pathway activation does not only enhance transcriptional signaling but also RAS signaling.

Similarly, phosphorylation can affect RAS activation, localization and effector interactions [80] (Figure 5E–G). SRC family kinases preferentially bind to the activated forms of all three RAS family members and phosphorylate them on tyrosines 32 and 64 located in the switch I and II regions, respectively [81,82] (Figure 5E). This diminishes the affinity for RAF kinases but increases binding of RAS-GAPs promoting RAS deactivation. The inhibitory tyrosine phosphorylation is reversed by the SHP-2/PTPN11 phosphatase, and pharmacological inhibition of SHP-2 suppresses ERK pathway activation and tumors in mouse glioblastoma models [83]. Interestingly, tyrosine phosphorylation of RAS proteins by ABL on tyrosine 137 has the opposite effect—it enhances RAF kinase binding to HRAS and downstream signaling [84] (Figure 5F). The ABL kinase is activated by HRAS in a RIN1 dependent manner [84], but RIN1 also stimulates HRAS sequestration in endosomes [72]. Thus, RIN1 may exert a complex regulation on RAS signaling acting as accelerator and brake-likely in this temporal sequence.

Phosphorylation of serine 181 in the KRAS polybasic domain by protein kinase C (PKC) relocates KRAS from the PM to mitochondria where it can interact with Bcl-XL and trigger apoptosis [85] (Figure 5G). However, this phosphorylation also enhances KRAS activation and transforming capabilities by decreasing its deactivation by GAPs [86]. Thus, this phosphorylation simultaneously changes subcellular localization, effector binding and biological outcomes.

There is also evidence that the modification of effector proteins can modulate RAS binding. The phosphorylation of CRAF on serine 43 by cAMP dependent kinase (PKA) or ERK can reduce RAS binding [87,88]. Likewise, the phosphorylation of serine 151 in BRAF by ERK was reported to reduce RAS binding [89], although this finding was disputed and may be cell type specific [90]. Both serine 43 and serine 151 are adjacent to the RAF RBD and their phosphorylation may directly influence the interaction interface. However, there are also more subtle mechanisms. For instance, PI3K is held in an inactive conformation by the regulatory p85 subunit preventing the activation of the p110 catalytic subunit [91]. Upon activation of receptor tyrosine kinases, the SH2 domains of the p85 subunit can bind to tyrosine phosphorylated residues in the receptors releasing the p110 RBD to bind to RAS. Similarly, the BRAF and CRAF kinases are held in an inactive conformation by 14-3-3 binding clamping the regulatory N-terminus together with the catalytic C-terminus [92,93,94]. Dephosphorylation of the N-terminal 14-3-3 binding site (serine 259 in CRAF and serine 365 in BRAF) is needed to allow RAS binding and activation [92,94]. In CRAF this critical dephosphorylation is enabled by the recruitment of PP1 or PP2A phosphatases that associate with CRAF at the membrane [95,96]. This phosphatase recruitment seems to be mediated by another RAS protein, MRAS, and the SHOC adapter protein, and this MRAS-SHOC-PP1 complex can dephosphorylate both CRAF serine 259 and the corresponding serine 365 in BRAF [97]. Such mechanisms, where PTMs of effectors regulate RAS binding, are likely to also be encountered in other effector pathways once they receive the detailed attention the RAF and PI3K pathways have.

## 3. Structural, Functional and Dynamic Organization of RAS Signaling Complexes

The formation of different RAS signaling complexes is also controlled by the biophysical properties of RAS proteins and their dynamic interactions with the membrane environment. As described in more detail in Section 3.3, the localization of RAS isoforms in different membranes is a key contributor to the regulation of RAS signaling [98,99]. Additionally, the traditional view that RAS GTP is the fully and only active conformation of RAS has been challenged by several lines of evidence that show that RAS aggregation and the different conformations RAS proteins can acquire in the membrane environment regulate the accessibility of their protein binding domains to their interactors [6,100,101]. Differences in aggregation and conformational status are also likely to contribute to the different affinities that different RAS family members show towards effectors interacting via highly conserved interaction domains [6]. In addition, dynamic structural effects can also occur in effectors and contribute to the binding characteristics of different RAS-effector complexes [100]. Here, we briefly summarize the possible roles of these higher order structural effects.

### 3.1. RAS Aggregates: Dimerization and Nanoclusters

Different lines of evidence show that RAS proteins can form aggregates in the membrane that seem necessary for the formation of some effector complexes. For instance, RAS dimer formation was proposed to be necessary for activating effectors that include a dimerization step in their activation, such as RAF proteins [102,103]. The existence of these dimers in vivo is still discussed in the literature, but in vitro experiments have shown that forced RAS dimerization increases the activation of the ERK pathway [104]. In support of the existence and relevance of RAS dimers in vivo, the expression of an oncogenic KRAS dimerization mutant in transgenic mice resulted in the loss of the tumor suppressor effect of wildtype KRAS, which is thought to be exerted by dimerization [105]. Extrapolating these data suggests that KRAS dimerization would be necessary for the activation of some effectors involved in transformation and that RAS dimerization is likely an important regulatory element that organizes RAS complexes in the PM.

Although there is still further work to be done to confirm the existence and relevance of RAS dimers in the membrane, there is wider agreement for the existence of RAS aggregates in the PM, which are named RAS nanoclusters. RAS nanoclusters have been proposed to be necessary for the activation of the ERK pathway by RAS, and calculations showed that ~40% activated RAS are part of these complexes [33]. All RAS isoforms can form nanoclusters, which are small (<20 nm), short lived (<1 s) and estimated to include 6–10 molecules of RAS [33,106,107]. Interestingly, results from a combination of biochemical experiments and computational modeling indicate that the activation of RAF by the RAS proteins only happens when RAS proteins are aggregated in nanoclusters [106]. The computational models indicate that the advantage of nanoclustering is to generate an extra layer of regulation that converts graded, analogue inputs from growth factor receptors into digital outputs of ERK activation. This is accomplished by the formation of RAS nanoclusters in proportion to the growth factor stimulus, and each nanocluster providing a fixed output of ERK activation [106]. Although most of these investigations studied the interaction of KRAS with CRAF, it is plausible that RAS nanoclusters also regulate the interaction with other effectors in the PM. For instance, an increase of RAS nanoclustering caused by an increase in BRAF/RAF1 dimerization upon treatment with BRAF inhibitors may prevent the interaction of RAS proteins with PI3K [108]. This mutual enhancement of dimerization and clustering between RAF and RAS proteins could preferentially allow the activation of pathways that include dimerization or oligomerization as activation steps at the expense of pathways that do not require the formation of multimeric protein complexes. It could also be a mechanism to promote the activation of low-affinity effectors as the formation of the nanocluster leads to a localized increase of concentration of RAS that can drive cooperative associations. Intriguingly, overexpression of mutant RAS proteins has been shown to favor the formation of nanoclusters [100], which could explain why some of the oncogenic mutants interact with effector proteins that do not bind to the wildtype RAS isoforms. In addition, RAS proteins can change the composition of membrane lipids through their membrane anchors acting as sorting devices [109,110]. This property is regulated by GDP/GTP binding and becomes especially pronounced when RAS proteins form nanoclusters that generate their own lipid environment in membranes. Intriguingly, nanoclusters containing different RAS isoforms can crosstalk via membrane lipids enabling, for instance, activated HRAS nanoclusters to disrupt the formation of KRAS nanoclusters [111]. The organization of RAS signaling units into nanoclusters is intimately linked to a finer level of structural organization that pertains to the interplay between RAS conformations and membrane components as discussed below.

### 3.2. Regulation of RAS Conformation by Membrane Interactions

Computational simulations of membrane bound RAS conformations validated by mutagenesis experiments indicate that RAS isoforms can acquire different conformational states in the PM (reviewed in [112]). These dynamic conformation states of RAS seem to be regulated by different posttranslational modifications that modulate the electrostatic interactions with the phospholipids mediated by the hypervariable region (HVR) that is located at the C-terminus of HRAS, KRAS and NRAS. However, in addition to the HVR lipid modifications (and the KRAS4B poly basic sequence), the catalytic domains also interact with membrane components including lipids and phosphate groups on the lipids. These interactions are mediated by selected residues that are regulated by the GTP/GDP activation status of the protein and by specific mutations associated with oncogenesis or rasopathies [113,114,115]. Although more work is necessary to fully characterize the dynamic conformation status of RAS, these studies indicate that HRAS, KRAS and NRAS can adopt distinct conformations in the PM when bound to GTP [101,115,116]. Interestingly, in some of these conformations the catalytic domain engages in PM interactions and is unavailable for effector binding suggesting that RAS proteins can adopt autoinhibited states in the membrane environment. Early evidence showed that the G-domain, i.e., the RAS catalytic domain, can adopt different orientations versus the PM, and that only one orientation allowed the interaction with the scaffold protein galectin-1 and the RAF and PI3K effectors [114]. Further studies to determine the characteristic conformational states of the different RAS isoforms indicate that in addition to the two conformational states originally identified, each isoform can adopt specific states that might be determined by the different lipid modifications that they have in the HVR. For instance, NRAS can acquire an additional conformational state when associated with a membrane [116]. The KRAS protein can exist in several conformational states (up to five in the case of the KRAS4A splice variant), but all studies seem to confirm that even in the GTP-bound form there is at least an active and inactive conformation that regulate the interaction with effectors [115,116]. Interestingly, membrane composition, which varies between PM microdomains and cell types, also impacts the conformation of RAS proteins [117]. This can further contribute to explaining the existence of specific interactors in different cell lines and differences in effector binding between oncogenic and wildtype RAS proteins. In summary, the mutual regulation between RAS conformation and nanoclustering and membrane composition provides a rich self-organizing environment that can determine effector interactions and signaling specificities in a versatile combinatorial way.

### 3.3. RAS Signaling from Different Subcellular Membranes

Reports from several groups have shown that all three RAS proteins could signal from different subcellular localization, and that this could result in the differential activation of effector pathways, e.g., ERK being mainly activation by RAS signaling from the PM, AKT from the PM and endoplasmic reticulum (ER), and RALGDS from the PM and Golgi apparatus [85,118,119,120,121,122,123,124]. These studies are highly relevant for drug discovery efforts, as many approaches to inhibit RAS have focused on displacing RAS from the PM [8,125]. To gauge the validity of these approaches, RAS signaling from endomembranes needs to be understood. Therefore, we undertook a systematic study analyzing HRASG12V signaling from different subcellular compartments (PM: lipid rafts and disordered membrane; ER; Golgi) combining interaction proteomics, phosphoproteomics, and transcriptomics using a new method for data integration [126]. This analysis recovered the known HRAS effector pathways and discovered new ones (Figure 6). Quantitative interaction proteomics showed that only 5% of HRASG12V interactions were common to all locations. Most of the protein interactions were found with HRASG12V located at the ER. However, most of the differentially induced phosphorylations were controlled by HRASG12V signaling from the PM. Very interestingly, most genes were controlled by HRASG12V signaling from the ER. However, transcription factors critical for transformation, such as MYC, RB1, STAT3, were targeted through multiple pathways from the PM pointing to RAS signaling from the PM as an important contributor to cell transformation. However, we also discovered new pathways emanating from other subcellular compartments that impinge on cell transformation. For instance, HRASG12V located at the Golgi produced a p53 mediated survival signal, and HRASG12V signaling from the ER controlled cell migration via an ERK dependent pathway. These results suggest that changing RAS subcellular localization could be a very selective tool to achieve certain effects.

## 4. RAS Isoform Specific Signaling

For the first two decades of RAS research the mainstream view was that all RAS isoforms were functionally redundant and shared the same interactors [6,99]. However, at the turn of the century evidence started accumulating that HRAS, KRAS and NRAS may have specific functions and different affinities for their main effectors RAF and PI3K or even isoform specificity for some interactors, such as RASSF1A and calmodulin [127,128]. Additionally, growing clinical evidence showed that the mutation frequency of the three RAS genes is very different, which may indicate that different oncogenic RAS isoforms may have different transforming effects [6], regulate different effectors and even may determine differential response to therapy [129]. Therefore, the current dominant view is that the 3 members of the RAS family have both redundant and specific functions. Although we still lack a full explanation how this specificity is determined, several studies have shown that the RAS isoforms might have different interactors. In this section, we review recent evidence obtained mainly by MS-based proteomics screening that shed some light of the existence of isoform specific interactomes.

### 4.1. The HRAS, KRAS and NRAS Interactomes: Current Knowledge

Current information available in public databases, such as Huri [35], BIOPLEX [130,131], and STRING [132], together with two recent publications that analyzed RAS-mediated complexes using AP-MS (affinity purification combined with mass spectrometry) [133] and BioID (proximity-dependent biotin labeling) [134], suggests a large number (1308) of interaction partners for the three isoforms with only 17% (389) shared interactors (Figure 7A). A functional analysis of RAS interaction partners using the SysGO database [36] shows that the ‘subcellular localization’ category is enriched for cell–cell junctions, endosomes, vesicles, cytosol, PM, actin microfilaments, and the Golgi apparatus (Figure 7B, left). With respect to the ‘protein function’ category, RAS interactors are enriched for cell junction & adhesion, signaling, neuronal system, cytoskeleton, organelles, and immune system & inflammation (Figure 7B, right). A more detailed analysis of all RAS interactors identifies 30 classical effectors (containing RBDs) and a large fraction of signaling-related proteins, such as proteins related to cell fate decisions, protein kinases and phosphatases, adaptors/scaffolds, and small GTPases and regulators (GEFs—guanine nucleotide exchange factors; GAPs—GTPase activating proteins) for the RAS (13 RasGEFs and 13 RasGAPs), Rho, Rab, and ARF superfamilies (Figure 7C). The RAS interactome also includes some previously described effectors for which domain no RBD was predicted using available domain prediction tools, such as Sin1 (MAPKAP1) [135].

This spectrum of interactors is far bigger than the bona fide RAS interactors we have studied so far, suggesting that we only have scratched at the surface of RAS functions. This conclusion is supported by the functional classifications (Figure 7C). Most previous studies have focused on the signaling role of RAS, and signaling comprises a large part of RAS functions. In particular, the regulation of phosphorylation networks stands out—RAS interacts with 65 protein kinases and 29 phosphatases, that is ~13% of the whole human kinome and ~14% of the phosphatase complement, respectively [136]. Another interesting feature is the cross-regulation of other G-proteins of the Rho, ARF, and Rab families indicating the existence of a G-protein network. The functional relevance of such crosstalk is documented by the synergy between RAS and Rho family proteins in transforming cells [137,138]. However, the large number of G-protein regulators in the RAS interactomes indicate much wider crosstalk. Much of this crosstalk seems to be dedicated to the regulation of membrane trafficking, cell adhesion, and the cytoskeleton, where several G-protein families are involved. Another big share of RAS functions pertains to the control of metabolism, which is moving into the limelight as providing new targets for RAS driven cancers [139]. Indeed, hexokinase 1 has just recently been shown to be directly regulated by KRAS [140]. Thus, RAS is truly a hub for the coordination of diverse cellular and biochemical processes, posing the question about the role of different RAS isoforms and different RAS activation states (GDP- vs. GTP-bound) in these regulatory networks. In the next section we review recent studies that have tried to disentangle RAS isoform specific functions by mapping RAS isoform specific interactomes.

### 4.2. Mass Spectrometry-Based Proteomics Reveals Differential RAS Interactomes

As RAS oscillates between GDP and GTP bound states, many of the functionally important binding partners may only interact transiently and hence escape detection by classic pulldown or co-immunoprecipitation techniques. One of the technologies to overcome this limitation is proximity-dependent biotin identification (BioID), where the bait protein is fused to a biotin ligase that covalently transfers biotin to other proteins within a 10–15 nm radius. The biotin-tagged proteins can be efficiently isolated using affinity purification with streptavidin beads and identified by mass spectrometry (MS) [141]. Using BioID to map the KRAS interactome in HEK293T cells identified 748 proteins [142], which interacted with wildtype KRAS, the oncogenic KRASG12D mutant, the dominant inhibitory KRASS17N mutant, which can be considered a GDP loaded inactive KRAS [143], and the KRASG12D/C185S mutant, which cannot be farnesylated and is therefore compromised in membrane binding [144]. Only 122 proteins were shared interactors between all four KRAS constructs, 116 proteins selectively bound to KRASG12D/C185S, and 133 proteins associated only with the membrane binding variants. The shared proteins were mainly involved in protein expression and trafficking suggesting that they belong to a core machinery rather than being RAS specific. Interestingly, the membrane resident KRAS proteins associated with other G-proteins and their regulators (GEFs and GAPs) as well as with receptor tyrosine kinases and other kinases involved in cell signaling. The large overlap between GTP- and GDP-loaded KRAS proteins shows that many interactors can bind to both active and inactive KRAS conformations, and that this simple concept of RAS being an on/off switch may need to be revised. The cytosolic KRASG12D/C185S specific interactors were enriched in proteins involved in nucleic acid binding and transcription. This may be an artefact, but could also indicate that KRAS may carry out functions in the nucleus. Expression of the KRASG12D mutant in a pancreatic cancer cell line with the same endogenous mutation led to the identification of 56 interacting proteins, mainly comprising cytoskeletal and PM proteins. The latter included signaling and cell adhesion proteins [142].

A study using BioID to compare the interactomes for the G12V oncogenic mutants of HRAS, KRAS4B and NRAS in HEK-HT cells showed a high degree of overlap between the 477 RAS binding proteins identified [145]. A small fraction of proteins was preferentially associated with a single RASG12V isoform with 44 proteins specific for NRASG12V, 75 specific for KRASG12V, and only 4 exclusively binding to HRASG12V. This result suggests that the different RAS isoforms have largely redundant functions, and—when compared with the above study—that interactor specificity between different RAS isoforms is less than that between different mutants of the same RAS isoform. Importantly, the authors used this information to perform a CRISPR-Cas9 screen to knock out 474 of the genes identified in this interactome in order to identify possible nodes that contribute to RAS oncogenesis. This analysis showed that while most of the targeted genes identified to have a role in RAS transformation were shared among the isoforms, 21% of the genes were enriched in cells transformed by just one of the RASV12 isoforms. Among these proteins the authors demonstrated that Phosphatidylinositol-4-Phosphate 5-Kinase Type 1 Alpha (PIP5K1A), the most negatively enriched gene in KRASG12V transformed cells, is a specific interactor of this isoform which mediates PI3K and ERK signal downstream of oncogenic KRAS. This work also showed that targeting PIP5K1A (which is a druggable target [146]) is potentially a new and selective avenue to treat mutant KRAS tumors. Knocking down PIP5K1A expression specifically reduced the viability of KRAS transformed cells versus NRAS or HRAS mutated cancer cells and had an additive effect with MEK inhibitors in KRAS mutant pancreatic cancer cell models. Thus, these results showed that identifying RAS isoform-specific interactors combined with functional genetic screens can successfully identify novel and highly selective therapeutic opportunities in mutant RAS tumors.

The results that the number of RAS isoform specific interactors is small compared to shared interactors was also observed in K-562 chromic myeloid leukemia (CML) cell lines [75]. This work showed that there is a shared core of 123 proteins that interacted with all of the RAS isoforms tested (HRAS, NRAS, KRAS4A and KRAS4B). However, this work also showed that all the isoforms have specific interactors. HRAS had only 6 isoform specific interactors (out of 153), NRAS had 28 (out of 232). Interestingly, the two KRAS splicing isoforms KRAS4A and KRAS4B had more specific interactors combined (72), but they also featured splice form specific interactions with KRAS4A specifically binding to 15 proteins and KRAS4B to 29 proteins. These two proteins have a very high degree of sequence homology and only differ in the sequence of their HVR, which must play a role in the regulation of selective interactions. The KRAS4A splice form is transforming, but poorly characterized as it is considered a minor splice variant compared to KRAS4B, which is commonly referred to as KRAS. However, recent data suggest that KRAS4A is widely expressed and of similar abundance as KRAS4B in many tumor cell lines [147], and that KRAS4A has stronger transforming abilities due to enhanced binding of the CRAF kinase [133]. Remarkably, the existence of specific interactors for both KRAS splicing isoforms was also shown in another study using affinity purification-mass spectrometry (AP-MS) with FLAG-KRAS constructs [133]. In this case, 103 and 94 proteins specifically interacted with KRAS4A and KRAS4B, respectively. Moreover, this study also compared both wildtype KRAS4A and KRAS4B splice forms with the respective G12D mutants. More than half of the interactors bound to both wildtype and mutant versions, including proteins involved in KRAS posttranslational processing. On the other hand, many known RAS effector proteins, such as RAF kinases only interacted with the mutant forms, showing that the wildtype and mutant isoforms also have specific interactors. These observations indicate that some of these mutant specific interactors may be good therapeutic targets that could block oncogenic KRAS signaling without affecting wildtype KRAS.

To account for possible interactome changes due to cell type-specific protein expression, a BioID screen of the interactomes of the wildtype HRAS, KRAS4B and NRAS proteins and the most commons mutations of these proteins, HRASG12V, KRASG12D and NRASQ61 was carried out in cancer cell lines where these mutations usually occur (HRAS, bladder; KRAS, colorectal cancer; NRAS, melanoma) [134]. As in previous studies, this work identified a group of common interactors (150), which was enriched for known RAS interactors. Interestingly, in this case HRAS showed the largest number of specific interactors (118) compared to 77 NRAS and 64 KRAS specific interactors. This observation highlights the importance of using relevant cell lines to identify the interactome of oncogenes in their ‘natural habitat’. To further characterize the role of these interactors in Ras functions, the authors preformed a CRISPR-based screen targeting a set of 130 proteins of newly identified interactors common to all RAS isoforms. This led to the identification of a new interaction between RAS proteins and mTORC2 that is necessary for cancer cell growth mediated by oncogenic RAS. mTORC2 is a multi-protein kinase complex that phosphorylates and activates other kinases involved in cell proliferation, survival and metabolic regulation, including AKT, protein kinase C (PKC) family members, and serum- and glucocorticoid-induced kinases 1 (SGK1) [148]. Since mTORC signaling is the object of intensive drug development, this finding further stresses the importance that characterizing RAS complexes can identify new potential targets for the treatment of RAS mutant tumors. Interestingly, a follow-up analysis showed that HRAS interacts with MET, KRAS with IFNGR1, and NRAS with ERBB2 and ephrin membrane receptors [149], supporting the idea that RAS isoforms function is regulated by different receptor systems in the PM. This functional analysis also indicated that KRAS may play role in PM organization since its interactome is enriched in PDZ domain proteins. HRAS interacts with several RAB GTPases and SNARES indicating that this isoform either regulates vesicle trafficking or itself undergoes extensive trafficking between the PM and endomembranes. The link between RAS and mTORC2 was also found in a study that mapped the interactomes of wildtype HRAS, KRAS4B and NRAS in Flp-In T-Rex 293 cells [150]. This study further identified over 800 interactors of wildtype RAS isoforms, showing that each isoform has specific interactors, i.e., HRAS (197), KRAS4B (95) and NRAS (81). It confirmed the regulation of pathways related to cell adhesion, cell junctions and vesicle trafficking, but also revealed a strong enrichment of proteins involved in oxidative phosphorylation for all RAS isoforms suggesting that RAS may directly regulate energy production.

Expanding the proximal RAS interactomes to also map components further downstream identified a network of interconnected G-proteins in lung cancer cells [151]. In this study, the authors used GDP and GTP locked GST-fusion proteins of HRAS, KRAS and NRAS to identify interactors in cell extracts. Interestingly, this analysis showed small differences between the GTP and GDP locked RAS proteins. However, when they expressed GFP-tagged RAS constructs in A549 lung adenocarcinoma cell lines, they observed marked differences, indicating that the intact cellular environment has a larger influence of interactor binding than the type of nucleotide loading. In this case, the HRAS interactome showed the greater number of specific interactors. Interestingly, RAF kinases (ARAF, BRAF, CRAF) bound mainly to oncogenic KRAS and NRAS, but not HRAS. Based on the proteomics results and a preliminary CRISPR screen, the authors designed a dual knockout CRISPR library where they could target 119 proteins containing strong KRAS interactors and proteins suspected to be involved in KRAS transformation of lung cancer cells. With this library they carried out a pairwise synthetic lethality CRISPR screen in two lung adenocarcinoma cell lines discovering 548 binary genetic interactions in A549 cells and 447 in H23 cells, of which only 59 overlapped. These values indicate that the cell line-specific variability of genetic interactions is considerable and in a similar range as variations in protein–protein interactions [152]. The analysis confirmed the importance of the RAS-RAF-MEK-ERK pathway, but also identified RAP1GDS1 and RHOA as a synthetic lethal gene pair that—when knocked down—specifically impaired the proliferation and tumorigenicity of mutant KRAS lung adenocarcinoma cells, while having minimal effect on wildtype KRAS cells. These studies show that the combination of proteomics and genetic screens seems a powerful method to discover promising molecular targets for new therapeutic interventions.

## 5. Conclusions and Future Perspective

RAS never ceases to surprise. That is the take-home message that summarizes the results of the last decade of RAS research and likely will be the leitmotif for the next 10 years. However, we are currently at a critical point in RAS research where we can see handles emerging that allow us to get to grips with taming oncogenic RAS. Therefore, we will discuss here the results described above mainly under this aspect. The evolving efforts to target RAS-driven cancers have been extensively reviewed [8,125,153,154,155]. Thus, we only discuss some general principles and how the studies about RAS signaling reviewed here may impinge on them.

The obvious and straightforward way to target RAS would be to design inhibitors that block the catalytic pocket that binds GTP/GDP. This concept has worked remarkably well for kinases, where a host of potent and specific inhibitors is available that slip into the ATP binding pocket and block catalysis. Unfortunately, the high affinity of GDP/GTP binding to RAS (picomolar) compared to the affinity of kinases to ATP (10–20 micromolar) renders this approach hopeless [8,154]. Our best kinase inhibitors have low nanomolar affinities and still require high micromolar concentrations to outcompete the ~1 mM concentration of ATP in the cell. It would require compounds of unheard femto-or attomolar affinities to outcompete the ~1 mM concentration of GTP in the cell. Apart from the GTP/GDP binding pocket, no other pockets were seen on the numerous molecular RAS structures to which a drug could bind. Therefore, RAS proteins were considered undruggable by a direct approach. This has changed now with the discovery of a pocket that dynamically opens up when RAS proteins cycle between inactive and active states [156,157,158]. Targeting this pocket can interrupt the conformational cycle and would have the advantage that much of the classic drug development machinery can easily be adapted to generate selective and high-affinity compounds. However, as many of the recent studies show, there likely is no strictly inactive or active RAS conformation [159,160,161]. Depending on the dynamic of the GDP/GTP exchange and the molecular microenvironment, RAS activation are 55 shades of grey and any interference targeting conformational states will need to carefully map out what signaling capabilities are associated with that state. This is a curse as much as an opportunity. It will require very detailed and comprehensive studies of RAS signaling in different subcellular and tissue contexts, but on the other hand may deliver highly selective drugs that block only transforming RAS functions.

A conceptually related but practically very different approach is to exploit peculiarities of different types of RAS mutations. Emerging evidence suggests that different RAS mutations orchestrate different ensembles of downstream signaling, and hence may open opportunities for very selective interference [162,163,164]. However, targeting opportunities may be limited. Recently, an elegant chemical approach has targeted inhibitors to KRASG12C mutants by designing inhibitors that covalently crosslink to the cysteine that replaces the normal glycine at this position [8,165,166,167]. These inhibitors have shown encouraging results in phase I clinical trials against KRASG12C lung cancer, but this mutation is rare in all other KRAS driven cancers.

The next step beyond targeting RAS itself is targeting its interaction with effector proteins. As the interaction surface is large and mainly hydrophilic, the development of direct interaction blockers is daunting. The progress made is still under debate. For instance, the chemical Rigosertib was reported to block the RAS–RAF interaction by binding to the RAF RBD [168]. However, this finding was disputed, showing that Rigosertib activates JNK-mediated stress signaling that inactivates RAF and RAS activation via phosphorylation-dependent negative feedback sites [169]. Thus, the jury is still out on the validity of these approaches. However, advances in the dovetailed understanding of structural protein features and functional effects will undoubtedly stimulate these efforts.

The largest efforts to target oncogenic RAS signaling were indirect, either targeting RAS effectors or RAS localization. The earliest efforts targeting the enzymes that modify RAS proteins to enable their membrane association proved ineffective in human clinical trials [2,8,154]. This is maybe a prime example of underestimating RAS complexity. Based on the assumption that all RAS proteins are equal, these compounds were developed using HRAS model systems, and failed completely in KRAS driven cancers which constitute the majority of human RAS driven malignancies. More recent efforts have revived this theme and have developed inhibitors that prevent KRAS membrane localization based on detailed dynamic studies of KRAS localization [170]. However, as many other proteins are modified by the same mechanisms, the caveat about specificity remains, and no detailed data are available on what RAS-dependent pathways are affected by these compounds. New efforts targeting regulators of RAS modifications, such as SHP-2, are being explored [80,83], but it remains to be seen how effective they may be in blocking mutant RAS proteins.

This leaves us with arguably the largest-scale efforts to target oncogenic RAS, which is targeting the effector pathways that mediate oncogenic transformation. Based on data from cellular and animal model systems, the RAF-ERK and PI3K-AKT pathways seemed to be the main RAS effectors that are required to transform cells [6,8]. Potent and selective inhibitors were developed, produced promising results in mouse models, and comprehensively failed in clinical trials. Disappointing as these results were, they stimulated further important investigations, which seem to validate the initial strategy with an important stipulation. We must attempt to understand RAS-dependent signaling on a global level and in a tissue-specific fashion rather than by focusing on cherry picked pathways in a few convenient-to-use cell lines. The recent results described in this review offer the required detail, but also highlight the enormous complexity of the task. They also show that sophisticated data analysis and especially data integration is key to identifying the subtle variations that can be successfully targeted. In our opinion the most promising advances to tackle RAS driven cancers can come from the computational modeling of RAS effector pathways in individual patients. Such models will allow us to simulate the effects of different treatments in silico before they are given to the patient. Importantly, this strategy can employ and repurpose existing drugs shortcutting the 10–17 years development time for a new drug. This is a strong argument for using computational modeling to analyze the intricacies of RAS signaling and design the best ways of therapeutic interference with drugs we have at hand.

## Figures and Tables

**Figure 1 biomolecules-11-00236-f001:**
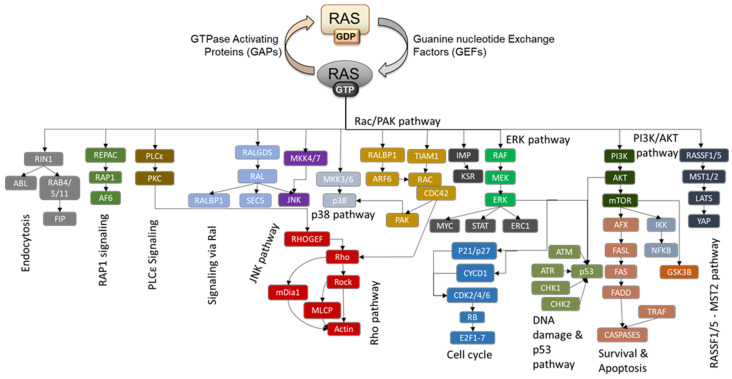
Known RAS effectors and signaling pathways. The RAS activation cycle comprises GTP binding aided by Guanine nucleotide Exchange Factors (GEFs), which puts RAS proteins into an active conformation where it can bind to effectors and activate downstream signaling pathways. The main and currently best-characterized RAS signaling pathways are shown. Effectors are released when GTP is hydrolyzed to GDP, and this intrinsic, but low, catalytic RAS activity is vastly accelerated by GTPase Activating Proteins (GAPs).

**Figure 2 biomolecules-11-00236-f002:**
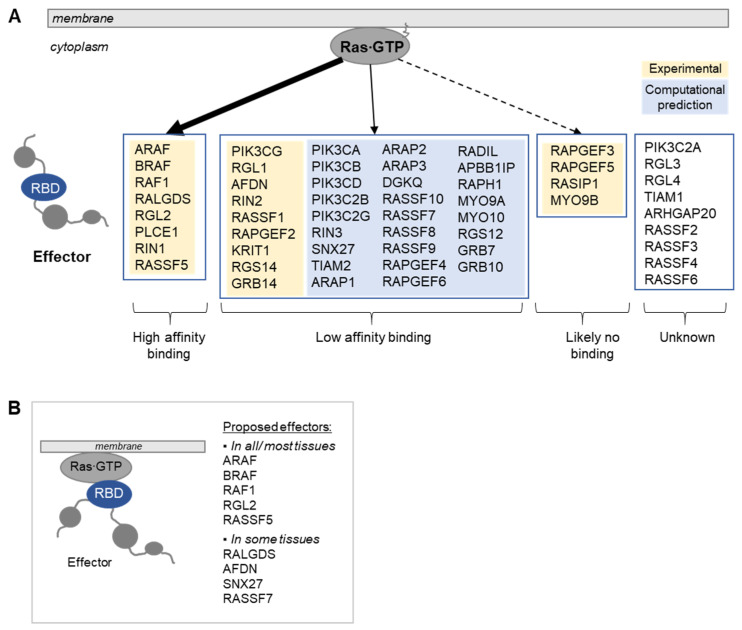
RAS and 56 classical effectors. (**A**) Classification of effectors based on the binding affinity of the RBD in complex with RAS∙GTP. High-affinity binders are defined by K_d_ values of ≤1 μM, and low-affinity binders by K_d_ values of >1 μM). (**B**) Schematic representation of the recruitment of effectors to RAS∙GTP via their RBD. Five effectors are predicted to be recruited efficiently to RAS∙GTP via their RBD only in all/most of the 29 human tissues, and four effectors are predicted to be recruited efficiently to RAS∙GTP via their RBD only in some of the 29 human tissues [23].

**Figure 3 biomolecules-11-00236-f003:**
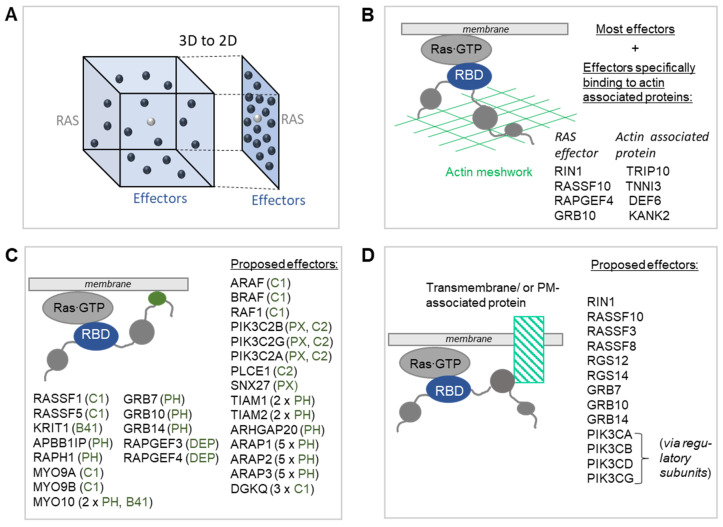
Mechanisms enhancing RAS effector recruitment. (**A**) The reduction of a three-dimensional space to the two-dimensions of a membrane can enhance the affinity between RAS and its effectors by >100 fold [29]. (**B**) Schematic representation of the recruitment of effectors to RAS∙GTP via their RBD and indication of the actin meshwork of the cell cortex. All effectors are predicted to increase their residence time in the PM once recruited via their RBD. Four effectors have been shown to bind to proteins that are located in the actin meshwork. (**C**) Representation of the recruitment of effectors to the membrane via their RBD and via membrane-binding domains. Twenty-six effectors contain domains that have the ability to bind to membranes (the names of the domains are indicated in green). (**D**) Recruitment of effectors via their RBD and via domains (or linear motifs) that bind to membrane-associated proteins.

**Figure 4 biomolecules-11-00236-f004:**
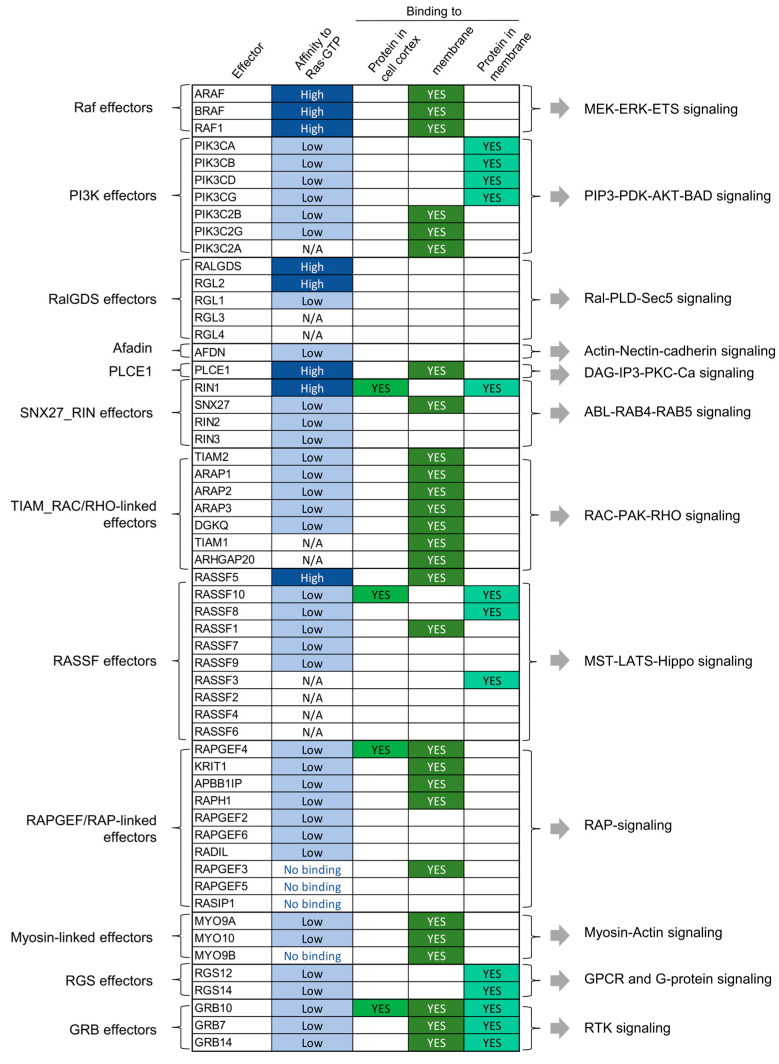
Summary of PM recruitment of 56 effectors grouped into 12 effector pathways. Classification of effectors into 12 effector pathways [17] and summary of the proposed membrane recruitment mechanisms (based on Figure 2 and Figure 3).

**Figure 5 biomolecules-11-00236-f005:**
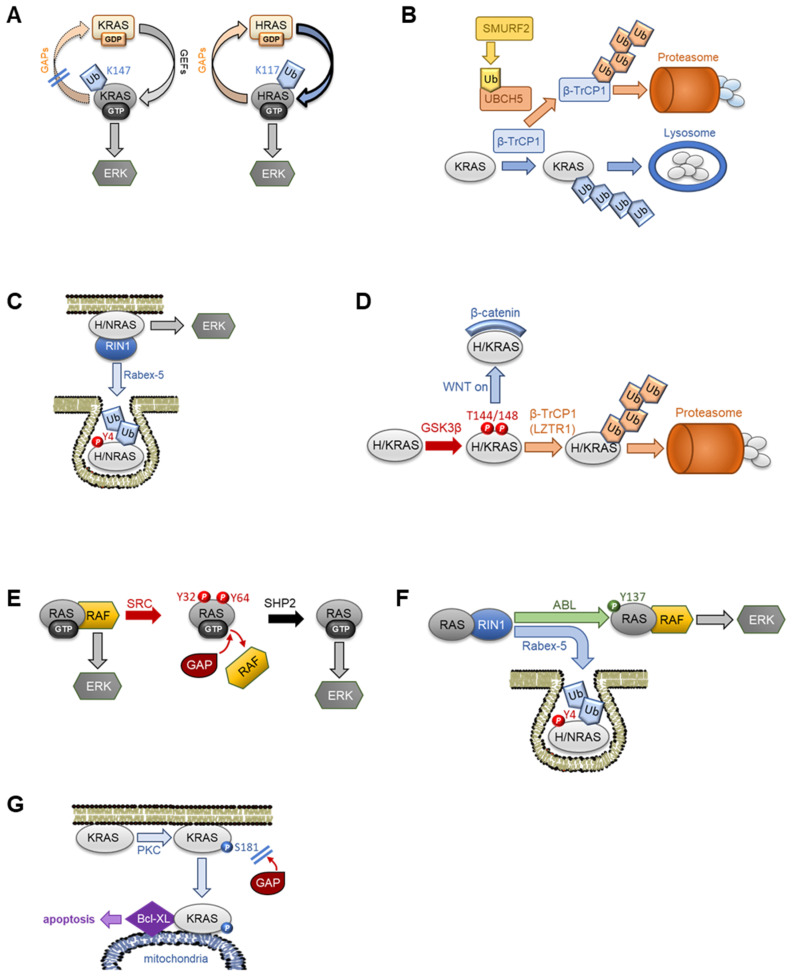
RAS regulation by posttranslational modifications. (**A**) RAS regulation by mono-ubiquitination. KRAS mono-ubiquitination at lysine 147 renders KRAS refractory to GAPs, while HRAS mono-ubiquitination at lysine 117 stimulates GEF independent GTP binding. Both mechanisms lead to the activation of downstream pathways such as ERK. (**B**) The constitutive lysosomal degradation of KRAS mediated by β-TrCP1 is impeded by Smurf2 mono-ubiquitinating UBCH5, which in turn poly ubiquinates β-TrCP1 and marks it for proteasomal degradation. (**C**) RAS regulation by mono-/di-ubiquitination. Rabex-5 can mono/di-ubiquitination NRAS and HRAS inducing endosomal localization and inability to activate ERK. (**D**) HRAS and KRAS degradation can be triggered by GSK3β phosphorylation of threonines 144 and 148, which induces poly ubiquitination by β-TrCP1. When WNT signaling is activated, stabilized β-catenin can bind to KRAS and physically shield the threonines 144/148 from phosphorylation resulting in KRAS protein accumulation. (**E**) Negative RAS regulation by tyrosine phosphorylation. The SRC kinase can phosphorylate all three RAS isoforms at tyrosines 32 and 64 leading to inhibition of effector binding and enhancement of GAP binding. These phosphorylations can be reversed by the SHP-2 phosphatase, which can restore signaling. (**F**) The RIN1 effector stimulates ABL mediated phosphorylation of Y137 which promotes RAF kinase binding and activation of the ERK pathway. However, RIN1 also stimulates Rabex-5-dependent ubiquitination of RAS, which leads to its endosomal sequestration. This ubiquitination is facilitated by the phosphorylation of tyrosine 4 by an unknown kinase. (**G**) PKC phosphorylation of KRAS at S181 has two effects. It reduces the interaction with GAPs prolonging KRAS signaling and induces the translocation of KRAS to the mitochondria, where it can bind to Bcl-XL and induce apoptosis.

**Figure 6 biomolecules-11-00236-f006:**
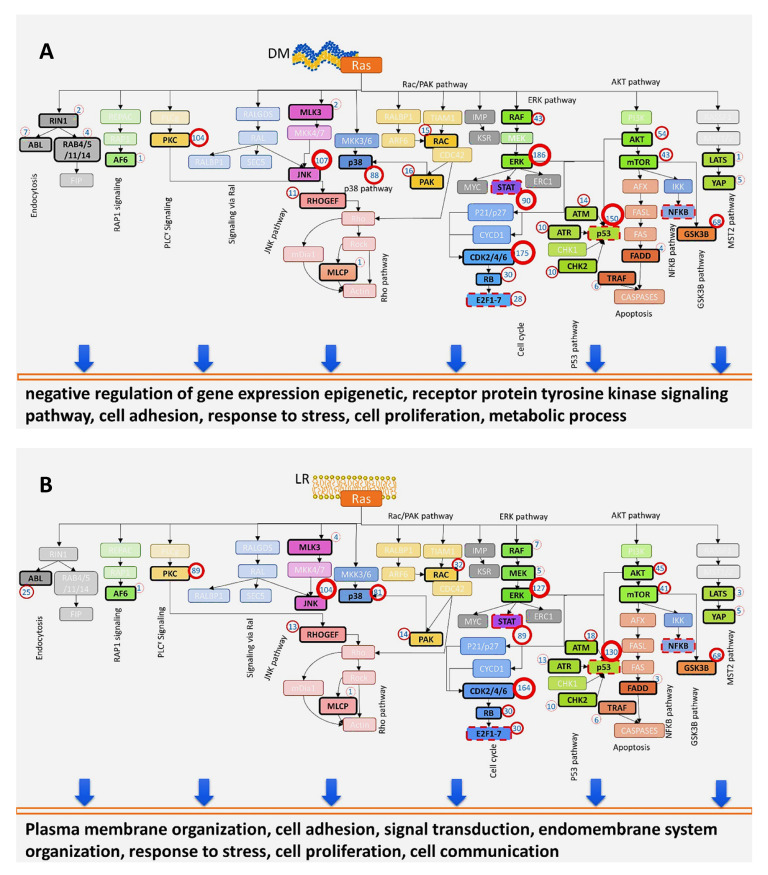
Signaling capacities of known RAS effector pathway components at different HRAS localizations superimposed on diagrams of these pathways. Signaling pathways activated when HRASV12 is located at the (**A**) disordered membrane (DM); (**B**) lipid rafts (LR); (**C**) endoplasmic reticulum (ER); and (**D**) Golgi apparatus (GA). The nodes displayed in bright colors are present in the corresponding network. The red circle adjacent to each node shows the number of interactions it participates in. In cases where a node represents more than one protein (isoforms or homologs), the signaling capacity of the node is represented by the total number of interactions involving all proteins corresponding to the node. The enriched GO terms of the HRAS localization specific transcriptome are shown at bottom of each pathway diagram. This figure is from the supplementary data of [126].

**Figure 7 biomolecules-11-00236-f007:**
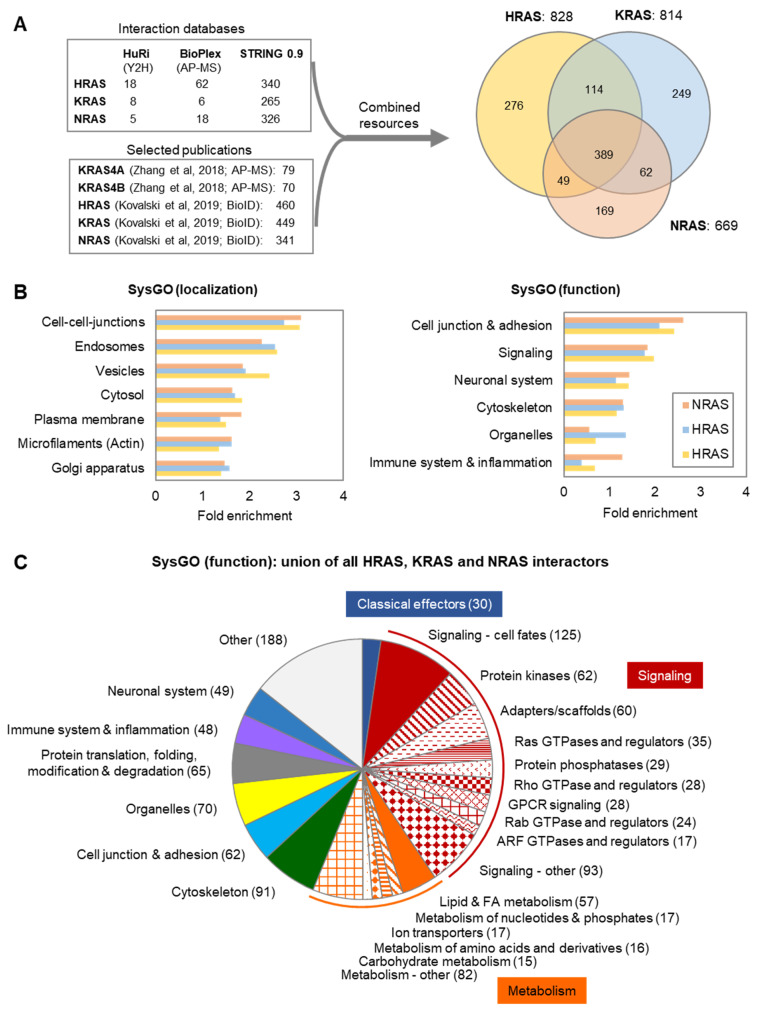
The RAS network as defined by interaction proteomics. (**A**) The RAS network obtained from databases and recent publications. Overview of interactions obtained from the HuRI, BIOPLEX, and STRING (highest confidence; 0.9) databases, and two recent publications [133,134] (left side). Venn diagram of interactions for HRAS, KRAS and NRAS (right side). (**B**) Enrichment of functional classes for HRAS, KRAS and NRAS interactors based on the SysGO database. (**C**) Functional analysis of all proteins in the union of HRAS, KRAS and NRAS interactors. Abbreviations: FA, fatty acids; Y2H, yeast two hybrid.

**Table 1 biomolecules-11-00236-t001:** RAS binding mechanisms of RAS effector groups.

Effectors	Mechanisms That Enhance Recruitment to RAS
1. RAF kinases	For the Raf family members it is well described that the C1 binds to the PM while at the same time the RBD interacts with RAS∙GTP [50]. Molecular dynamic simulations recently showed how the C1 domain reduces the fluctuations of RAS and RAF, and how this increases the population of RAS-Raf complexes at the PM and further enhances the already high affinity [51].
2. PI3Ks	The affinities of the RBDs of PI3Ks are low (mid to high micromolar range) suggesting that RAS∙GTP alone may not be sufficient to recruit members of this family [23]. In the case of class I family members, PM recruitment involves the regulatory subunits (PIK3R1-6), which bind to transmembrane receptors and adaptors [49]. For the class II PI3K family member PIK3C2A structural studies together with biophysical characterization revealed cooperative binding of the PX and adjacent C2 domains to different phosphoinositides (Kd values ranging from 4.5 to 52 μM) [52].
3. RALGDS	The group of RALGDS/RALGDS-like proteins does not contain domains for PM localization or interaction partners localized at the PM. Only two members (RALGDS and RGL2) bind with high affinity to RAS∙GTP using their RBDs.
4. Afadin	The RBD of AFDN binds to RAS∙GTP with weak affinity (3.03 μM), and AFDN does not contain domains for PM localization or interaction partners localized at the PM. Given AFDN’s role in cell adhesion its adjuvant binding domains seem to play a greater role localizing AFDN to adhesion proteins (e.g., NECTIN3 [53]) rather than to the PM.
5. PLCE1	While this phospholipase contains a C2 domain that could direct the protein to the PM, a recent 3D structure of (almost) full-length PLCE1 provided insights into its mechanisms and PM localization [54]. The model proposed that the C2 domain was not used for PM binding, but rather the αX–Y helix in the lipase domain. This orientation would enable the high-affinity RBD to bind to RAS∙GTP at the same time.
6. SNX27_RIN linked effectors	The RIN family member RIN1 has a high affinity of its RBD in complex with RAS∙GTP and can also bind to PM-associated proteins. SNX27 belongs to the family of sorting nexin proteins, which all contain PX domains [55]. Indeed, protein–lipid interactions mediated by the PX domain provide a means of docking to the PM or endosome, critical for the role of nexins in endocytosis and protein transport [56].
7. TIAM_RAC/RHO linkedeffectors	Members of this effector group have weak binding affinities of their RBDs in complex with RAS∙GTP. However, several members contain PH domains that may be used for PM recruitment after phosphoinositides have been produced.
8. RASSF linked effectors	Out of all ten RASSF family members, only RASSF5 binds with high affinity to RAS∙GTP via its RBD, and RASSF1 and RASSF5 additionally contain a C1 domain. RASSF1A is a bona fide effector of KRAS mediating its proapoptotic function [57,58]. For the mouse homolog of the effector RASSF5 (Nore1) it was demonstrated that in the inactive state the C1 domain packs against the RBD, and that binding of the RBD to RAS∙GTP displaces the C1 domain, which then exposes a lipid binding interface [59].
9. RAPGEF/RAP-linkedeffectors	No high-affinity binders are found for members of this effector group, but many contain domains with the ability to bind the PM. The DEP domain of RAPGEF3 can bind to phosphatidic acid at the PM with high affinity, but additional activation by cAMP and associated release of autoinhibition is required to enable interaction of the DEP domain with the PM [60].
10. Myosin-linked effectors	The myosin family effectors MYO9A, MYO9B, and MYO10 all contain domains (PH, C1, B41) that can bind to the PM. However, given the role of these effectors connecting cell junctions on one end to the actin filaments on the other hand, these effectors may rather be recruited to RAS proteins at cell junctions instead of the PM.
11. RGS effectors	The RGS effectors (RGS12 and RGS14) have weak binding affinities in complex with RAS∙GTP in complex with their RBD domain. RGS14 is a GTPase activating protein specific for the GPCR Gαi/o [61,62]. Both RGS12 and RGS14 effectors contain GoLoco motifs that bind specifically inactive Gαi1-GDP and Gαi3-GDP subunits, which promotes their translocation to the PM [63].
12. GRB effectors	The RBD domains of GRB7, GRB10 and GRB14 all display weak binding affinities to RAS∙GTP. However, all GRB effectors contain PH domains that have binding affinities (K_d_ values) in complex with phosphoinositides in the range of 4–10 μM [64]. The crystal structure of RASV12 in complex with the region of GRB14 spanning the RBD and PH domains suggest that GRB14 can simultaneously bind to RAS proteins via its RBD and to the PM via its PH domain [65].

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
