# Peer review of "The Ins and Outs of RAS Effector Complexes"

_biomolecules, 2021, doi:10.3390/biom11020236_

Round 1

Reviewer 1 Report

This is a very well-written comprehensive review on RAS effector and interactome based on the recent progress made using molecular studies, large scale omics studies, and computational modeling studies. This review provides new insights into the RAS signaling and would be beneficial to the RAS community.

I recommend its publication and have a few minor suggestions that authors may wish to incorporate in their revised manuscript:

  • It would be helpful if the authors comment if all the RAS interactome discussed in the second half of the review are likely to be effector proteins or may also include RasGEFs and RasGAPs. 
  • Although authors have focussed their review on key effectors, it would have been helpful to mention briefly some of recent RAS effector proteins that have appeared in the literature, such as hexokinase, Sin1 (part of mTORC2), etc.
  • A brief text on similarities and differences, if any, between RAS-associating (RA) and RAS-binding domain (RBD) would be helpful. It will also be useful if authors can comment on why most effector RBDs (despite containing similar ubiquitin fold as high-affinity effector RBDs) bind to RAS with weak affinity.
  • There are punctuation errors at many places; especially comma is missing in many places. A few examples are in these lines: lines 388, 455, 591, 657, etc.

Author Response

Point-by-point response to reviewer comments

Reviewer 1

This is a very well-written comprehensive review on RAS effector and interactome based on the recent progress made using molecular studies, large scale omics studies, and computational modeling studies. This review provides new insights into the RAS signaling and would be beneficial to the RAS community.

I recommend its publication and have a few minor suggestions that authors may wish to incorporate in their revised manuscript:

Comment 1: “It would be helpful if the authors comment if all the RAS interactome discussed in the second half of the review are likely to be effector proteins or may also include RasGEFs and RasGAPs.

Reply 1: The Ras interactome (1308 interactors) contains 30 effectors that contain RBDs, but also 13 GEFs and 13 GAPs for RAS (in addition to GEFs and GAPs for other RAS subfamilies (e.g. Rab, Rho, Arf). Part of this was mentioned already in the text, but we made the proportion of effectors vs GEFs/GAPs clearer now by adding the actual numbers to the text.

Comment 2: “Although authors have focussed their review on key effectors, it would have been helpful to mention briefly some of recent RAS effector proteins that have appeared in the literature, such as hexokinase, Sin1 (part of mTORC2), etc.”

Reply 2: We mention these effectors and the relevant publications now in the section about the Ras interactome. Indeed Sin1 (MAPKAP1) is part of the Ras interactome.

Comment 3: “A brief text on similarities and differences, if any, between RAS-associating (RA) and RAS-binding domain (RBD) would be helpful. It will also be useful if authors can comment on why most effector RBDs (despite containing similar ubiquitin fold as high-affinity effector RBDs) bind to RAS with weak affinity.”

Reply 3: We explain this now in chapter 2.2.

Comment 4: “There are punctuation errors at many places; especially comma is missing in many places. A few examples are in these lines: lines 388, 455, 591, 657, etc.”

Reply 4: We have corrected these mistakes and the lines mentioned and at other places.

Reviewer 2 Report

This is a very interesting and complete review about  RAS oncogenes the most commonly mutated protein in human cancers.

Nevertheless ras genes are implicated also in Noonan and Noonan-like syndromes also known as rasopathies and in a fully comprehensive review regarding ras genes, genetic syndromes due to autosomal inheritance of ras gene mutations  it should at least be  remembered.

Author Response

Reviewer 2

This is a very interesting and complete review about RAS oncogenes the most commonly mutated protein in human cancers.”

Comment 5: “Nevertheless ras genes are implicated also in Noonan and Noonan-like syndromes also known as rasopathies and in a fully comprehensive review regarding ras genes, genetic syndromes due to autosomal inheritance of ras gene mutations it should at least be remembered.

Reply 5: We mention RASopathies and relevant publications in the revised text.